# Olive Tree Belowground Microbiota: Plant Growth-Promoting Bacteria and Fungi

**DOI:** 10.3390/plants13131848

**Published:** 2024-07-05

**Authors:** Maria Celeste Dias, Sónia Silva, Cristina Galhano, Paula Lorenzo

**Affiliations:** 1Associate Laboratory TERRA, Center for Functional Ecology, Department of Life Sciences, University of Coimbra, Calçada Martim de Freitas, 3000-456 Coimbra, Portugal; paulalorenzo@uc.pt; 2LAQV-REQUIMTE, Department of Chemistry, University of Aveiro, Campus Universitário de Santiago, 3810-193 Aveiro, Portugal; soniasilva@ua.pt; 3Polytechnic Institute of Coimbra, Coimbra Agriculture School, Bencanta, 3045-601 Coimbra, Portugal; cicgalhano@esac.pt

**Keywords:** arbuscular mycorrhiza fungi, climate change, *Olea europaea* L., plant growth-promoting bacteria and fungi, sustainable agriculture

## Abstract

The olive tree is one of the most significant crops in the Mediterranean region. Its remarkable adaptability to various environments has facilitated olive cultivation across diverse regions and agricultural scenarios. The rising global demand for olive products, coupled with climate challenges, is driving changes in cultivation methods. These changes are altering the traditional landscape and may potentially reshape the structure and composition of orchard microbial communities, which can impact productivity and stress tolerance. Bacterial and fungal communities naturally associated with plants have long been recognized as crucial for plant growth and health, serving as a vital component of sustainable agriculture. In this review, we aim to highlight the significance of olive cultivation and the impact of abiotic stresses. We update the current knowledge on the profiles of rhizosphere and root fungal and bacterial communities in olive orchards and examine how (a)biotic factors influence these communities. Additionally, we explore the potential of plant growth-promoting bacteria and fungi in enhancing olive physiological performance and stress tolerance. We identify knowledge gaps and emphasize the need for implementing new strategies. A comprehensive understanding of olive-associated microbiota will aid in developing sustainable agronomic practices to address climatic challenges and meet the growing demand for olive products.

## 1. Introduction

Agriculture is facing multiple challenges due to the high pressure to produce more food to feed the growing world population, the necessity to adopt more efficient and sustainable systems, and the need to adapt to climate change [1,2]. Climate change, particularly extreme weather events, has impacted agriculture, mostly by reducing crop yield and productivity, leading to economic losses and increasing labor and equipment costs [2]. These weather events are projected to worsen, as climate models highlight increases in warming and precipitation variability, leading to more periods of extreme precipitation and drought [3]. In turn, to ensure that the food demand for the increasing global population is met under climate change constraints and to increase profitability, agrochemicals and irrigation have been overused, resulting in significant negative impacts on environmental quality and soil health [2,4]. In this scenario, there is an urgent need for the establishment of measures, starting, for example, with the implementation of fairer and environmentally friendly food systems [5,6]. In this context, the European Union (EU), through the Farm to Fork strategy, aims to accelerate the transition to more sustainable food systems with a lower environmental impact, increase crop resilience and adaptation, ensure food security and public health, and preserve the affordability of food [7,8,9]. Moreover, this strategy aims to decrease nutrient loss by around 50% and reduce fertilizer use by up to 20%, by 2030 [9]. Nevertheless, global pesticide use increased by 62% between 2000 and 2021, reaching 3.5 million tonnes in 2021. Most of this increase occurred between 2000 and 2016, with a small decline being observed until 2018, followed by renewed growth. Regarding fertilizers, the total agricultural use of inorganic fertilizers (measured as the sum of nitrogen (N), phosphorus (P_2_O_5_), and potassium (K_2_O)) was 195 million tonnes in 2021. This represents an overall increase of 60 million tonnes, or 44% compared with that in 2000. Fertilizer use increased in all regions between 2000 and 2021 [8,9].

In the past decade, efforts have been made to optimize agriculture productivity towards more sustainable management and to meet the EU goals [7,8]. The potential of microbial communities, particularly soil microbiota, to increase plant growth, development, and fitness, as well as soil health and fertility, has been described as a key role in the development of sustainable agriculture [10,11]. In fact, the biotransformation and degradation of xenobiotic compounds and contaminants such as metals, pesticides, and solvents within soil also take place in natural environments, and are mainly carried out by native heterotrophic soil bacteria (e.g., the genera *Pseudomonas*, *Micrococcus*, *Streptomyces*, *Corynebacterium*, and *Thiobacillus*) and most wood-degrading fungi (e.g., white-rots, such as *Phanerochaete chrysosporium*, and *Trametes versicolour*) [10,11]. Soil microbiota can offer some benefits to plants, including improvements in growth, nitrogen use efficiency, and biotic and abiotic stress tolerance [12].

Several classes of soil microorganisms have been described to stimulate plant performance, with plant growth-promoting fungi (PGPF) and plant-growth promoting bacteria (PGPB) being the most studied [7,13,14]. Rhizosphere fungi belong to the genera *Trichoderma*, *Penicillium*, *Phoma*, *Aspergillus*, *Fusarium*, and arbuscular mycorrhiza fungi (AMF), and have been described to provide a variety of benefits to host plants [15]. AMF are some of the most widespread soil microorganisms (e.g., phylum *Glomeromycota*) that colonize root plants [16,17]. Arbuscular mycorrhizal symbiosis with plant roots (where the mycelium colonizes the roots) involves the transfer of some plant photosynthates in exchange for mineral nutrients, particularly nitrogen and phosphorus, with the extraradical hyphae serving as an extension of the root system to facilitate nutrient uptake [17]. The PGPF mechanism of action involves improvements in soil aggregation, photosynthesis, phytohormone synthesis, volatile organic compounds, antioxidants, and nutrient uptake [15,18,19]. PGPB are groups of bacteria that can exist as free-living bacteria in soil or that colonize plant roots (rhizosphere bacteria can live around the roots or become established within the tissues of the host plant) [20]. These groups belong to the phylum *Proteobacteria* (e.g., the genera *Achromobacter*, *Azospirillum*, *Azotobacter*, *Acinetobacter*, *Burkholderia*, *Enterobacter*, *Pantoea*, *Psychrobacter*, *Pseudomonas*, *Rhizobium*, *Rahnella*, and *Serratia*) and the phylum *Firmicutes* (e.g., *Bacillus* and *Staphylococcus*), and are described to promote growth in many plant species [14,18,21]. Bacteria are able to grow rapidly and are by far the most abundant organisms in soils. Unlike eukaryotes, bacteria can be found in a wide range of environmental, chemical, and physical conditions including extremes of pH, temperature, and salinity. Being necessary for environmental sustainability, bacteria play important roles in the plant–soil system, firstly by both fixing and transforming nutrients vital to other organisms, and secondly by influencing the overall ecology of the system through positive or negative biotic interactions with other organisms. Indeed, PGPB can induce direct mechanisms, such as adjustments to hormonal signaling, nitrogen fixation, and phosphorus solubilization, as well as indirect mechanisms like systemic resistance, antibiotic production, osmotic adjustment, siderophore production, quorum quenching, and cell wall degrading enzymes [18,19]. The use of AMF and/or PGPB is becoming a promising strategy to reduce the negative effects of synthetic agrochemicals and move towards a more sustainable agricultural system, ensuring the sustainability of agriculture [18].

The olive tree (*Olea europaea* L.) is one of the most important crops in the Mediterranean region. This region produces around 68% of the world’s olive oil, with Spain, Greece, Italy, and Portugal being the main producers in Europe [22]. In recent decades, increasing interest and consumption of olive products has promoted the expansion of olive culture to countries like China, India, Australia, Brazil, Chile, Argentina, and South Africa [23]. To satisfy the globally increasing demand for olive oil, new olive orchards were established in the Mediterranean regions, and cultivation methods were significantly changed, from traditional dry farming to high-density systems with high inputs of agrochemicals and irrigation [24]. This new concept of olive orchards is incompatible with the scenario of climate change and water scarcity. Thus, the use of bio-based methodologies can be both practical and environmentally friendly, contributing simultaneously to enhancing plant tolerance to stress. In this review, we emphasize the importance of olive culture and the impact of abiotic stresses, and explore the diversity of rhizosphere and root endosphere microorganisms (PGPB and PGPF) associated with olive orchards, as well as their benefits and potential to promote olive physiological performance and stress tolerance. Moreover, some knowledge gaps are identified, and recommendations for new strategies are proposed.

## 2. Olive Culture and Climate Change Challenges

### 2.1. Olive Culture

*Olea europaea* is native to the coastal regions of the Levant, Anatolia, Greece, Sicily, and Italy, and of the Iberian Peninsula, as evidenced by the detection of olive pollen during the early Holocene [25]. According to Langgut et al. [25], the beginning of olive horticulture can be traced back to around 4800 B. C. in the northern Levant, which then spread through to the Mediterranean Basin. Today, olive farming is practiced in various continents [26]. In 2022, the olive orchards in the EU covered nearly 5 M ha, with Spain contributing to more than 2.6 M ha [27]. Nevertheless, olive cultivation has evolved over time, leading to the existence of two main types of olive orchards in the Mediterranean regions: the traditional system and the high-density system with intensive and super-intensive cultivation. Traditional groves still dominate cultivation systems with low plant density (less than 200 tress/ha), low mechanization, reduced fertilization and irrigation, and high productivity per tree but low productivity per hectare [7,22]. In contrast, the last three decades have witnessed a trend of intensification in olive cultivation, marked by the emergence of intensive (~200–450 trees/ha) and super-intensive (~1000–2500 trees/ha) irrigated, fertilized, and mechanized orchards that cultivate small-sized but highly productive trees [22,28]. This expansion promoted regional development and holds significant socio-economic value in terms of revenue for many farmers in the Mediterranean Basin, particularly in Spain, which stands as the major world producer [28]. 

In the EU, Spain is expected to produce around 765,000 tonnes of olive oil in 2023/24, Italy is expected to produce 280,000 tonnes, Greece is expected to produce 260,000 to9nnes, and Portugal is expected to produce 150,000 tonnes. Among the non-EU countries, the major producers are Turkey, Tunisia, and Marocco, these countries being estimated to have produced between 100,000 and 200,000 tonnes of olive oil each in 2023/24. Nevertheless, these values are substantially below the maximum already achieved, following the marketing year 2022/23, with the lowest production level in the EU since 1994/95. This below-average production, which pushed olive oil prices up throughout 2023, can be attributed to drought and extreme weather events in the south of the EU throughout 2022 and the spring and summer of 2023, highlighting the vulnerability of olive groves and olive oil production to environmental factors [29].

### 2.2. Effects of the Most Important Abiotic Stress on Olive Culture

The sustainability of agricultural systems, such as olive orchards, encompass factors of multiple dimensions, such as economic, social, and environmental [30], which, in turn, can be affected by a multitude of threats. From an environmental perspective, heat waves, dry periods, intense rainfall in a short time, soil erosion, and soil contamination by pesticides and chemical fertilizers can jeopardize olive production, thus negatively impacting the economic and social benefits of olive culture and its overall sustainability. In line with this, a number of different studies have reported on the impairments that abiotic stresses like drought, salinity, temperature, light, and contaminants have caused in olive physiology, metabolome, and/or microbiome [31,32,33,34,35,36]. 

The Mediterranean Basin is particularly susceptible to climate change, being already exposed to higher temperatures and alterations in precipitation patterns. Furthermore, predictions point out that in 2050, in the Mediterranean, there will be an increase in temperature of 0.8–2.3 °C and a decrease in rainfall of up to 200 mm per year, increasing the evapotranspiration of vegetation and atmospheric CO_2_ emissions [30]. Thus, despite the ability of olive trees to be well adapted to water limitations and high temperatures, it is forecast that crop yields will be affected, as already observed in 2022/2023 and 2023/24 in the EU. In line with this, it has been reported that drought stress on olive can impair several physiological parameters such as water content [37,38], electron transfer rates throughout the photosynthetic apparatus [39], carbon assimilation [38,40,41], redox homeostasis [42], nutrient uptake, and biomass accumulation [38]. Furthermore, drought can also alter the content and profile of olive fruit metabolites, including soluble sugars, unsaturated fatty acids, organic acids, and polyphenols, which in turn will impact the composition and quality of the oil [43,44]. Nevertheless, the impact of drought on olive’s physiology and biochemistry and even on plant survival is highly dependent on the cultivar, stage of development, and severity of stress [35,37,38,39]. Similarly to drought, heat stress can also induce oxidative stress, impair photosynthesis by changing the efficiency of photosystem II and electron transfer, or impair photosynthetic CO_2_ transport due to stomatal restrictions [45,46]. During oil accumulation in fruits, high temperatures can inhibit the expression of some genes involved in the olive oil biosynthesis pathway [47], thus reducing olive oil yield and quality [48]. Moreover, elevated temperatures have a significant effect on the reproductive growth and development of olives, including at the flowering stage, anthesis, pollination, and fruit set. Thus, not only are heat shock waves during oil accumulation of concern, but so are warm temperatures during the reproductive phase in winter [49]. As drought and heat stress frequently occur together, some studies have explored the combined effect of these stresses on olive. Some of the findings indicate greater detrimental impacts from combined stresses than from isolated ones, including lower photochemistry, superior oxidative stress, and membrane damage [45]. Combined stresses also alter the metabolite profile and richness of bioactive compounds, despite them being cultivar-dependent [50].

Groundwater quality is decreasing due to its overexploitation, leading to sea intrusion in aquifers and crop irrigation with salty water in the Mediterranean area [51]. Despite olive being considered moderately tolerant to salinity, this tolerance is dependent on the cultivar and can impact olive culture. For example, salt stress can reduce growth and biomass production [51,52], net photosynthesis and chlorophyll content [53], and water content [52], as well as cause a decrease in fruit weight and oil content [54]. Additionally, there have also been reports of membrane damage, leaf necrosis, leaf abscission, and plant mortality induced by salt stress exposure [52]. On the other hand, Trabelsi et al. [55] showed that saline water irrigation reduced the impact of drought on olive and facilitated physiological recovery after re-watering, highlighting that controlled watering with salty water may be a strategy against severe drought. 

## 3. Belowground-Associated Microbial Communities

### 3.1. Olive Rhizosphere and Root Endosphere Microbiota and Their Beneficial Effects 

Plant roots can select some of the microorganisms that inhabit the surrounding soil [2]. Plants release chemical signals (e.g., volatile organic compounds and root exudates) that act as a communication system with the environment, attracting beneficial organisms or activating defense mechanisms [56]. Root architecture and morphology, tissue growth stage, and plant genotype, as well as soil environmental conditions and agricultural practices, also play a role in the community and functioning of the microorganisms in the rhizosphere [24,57]. 

Olive orchards are good reservoirs of microbes, and several fungi and bacteria communities have been described in different plant compartments/organs, like the root endosphere, rhizosphere, fruits, leaves, and xylem [24,58,59]. These microbial communities can influence olive growth and productivity, and even help in the control of some pests and diseases [59]. However, several factors, like cultivar/genotype, season, environmental conditions, soil characteristics, and management practices, can impact microbial community assembly [20,24,59]. For instance, the implementation of high-density orchards with high use of synthetic agrochemicals, excessive tillage, and clearing affects soil characteristics (e.g., compaction, nutrients, and water content) and modifies the soil microbial communities [60,61]. 

Most of the research on belowground microbiota has been conducted in countries of the Mediterranean where olive culture is dominant, such as Spain, Italy, Portugal, Greece, and Tunisia, using methods and forms of analysis that allow for a wide description of the microbe communities. According to these studies, the rhizosphere and root endosphere of olive are mostly dominated by the bacteria phyla *Actinobacteria*, *Proteobacteria*, *Acidobacteria*, *Fermicutes*, *Bacteroidetes*, and *Verrucomicrobia*, and by fungal communities belonging to the phyla Ascomycota, Basidiomycota, and Glomeromycota (Table 1). Some of the genera and species identified in soil and roots (Table 1) are described as beneficial microbes (e.g., species belonging to the genera *Actinophytocola*, *Pseudomonas*, *Bacillus*, and *Glomus*), and others are considered deleterious microorganisms, such as in the case of the pathogenic fungi *Macrophomina phaseolina*, *Phomopsis columnaris*, *Fusarium oxyporum*, *Colletotrichum*, and *Botrytis*, and the bacterium *Erwinia* [58,59,62,63]. 

Regarding the beneficial effects of microbial communities (Table 2), some species of *Pseudomonas* and *Bacillus* are described to have an antagonistic effect on *Verticillium dahliae*, inhibiting the growth of this fungus, and improving plant growth through the increase in phosphate and nitrogen solubilization [57,62,64]. Additionally, many species of the genera *Streptomyces* and *Cladophialophora* are described as biocontrol agents for different plant pathogens (e.g., *Magnaporthe oryzae* and *Plectosporium tabacinum*), and the genus *Aureobasidium* suppresses anthracnose in olive trees [62,65,66]. Some species of the genera *Diaporthe* and *Preussia* have been described to produce secondary metabolites with antimicrobial activity that help to control the development of some plant pathogens [66]. *Cyanobacterium* spp. promote plant growth and can act as plant biocontrol agents [57]. Several bacteria from the phylum *Actinobacteria* (e.g., *Streptomyces* spp.) are described to promote plant growth traits by increasing ammonia and indole-3-acetic acid, phosphate solubilization, and siderophore production (high-affinity ferric iron chelator that binds with Fe^3+^), and the production of secondary metabolites like polyene macrolides, actinomycins, aminoglycosides, streptothricins, anthracyclines, cyclopolylactones, and quinoxaline peptides [67]. Some bacteria from the classes *Thermoleophilia*, *Bacilli*, and *Chloroflexi* and the genera *Micrococcus* and *Steroidobacter* are known to have a key role in olive adaptation to environmental stresses, particularly to arid and dry conditions [62,68,69]. Several bacteria from the phyla *Acidobacteria* (classes *Acidimicrobiia*, *Blastocatellia* and *Rubrobacteria*), *Planctomycetota* (class *Phycisphaerae*), *Verrucomicrobia* (class *Verrucomicrobiae*), *Proteobacteria* (class *Gammaproteobacteria*), and *Gemmatimonadetes* are very important for the quality and fertility of soils, since they are related to organic matter decomposition and degradation, and plant protection against biotic and abiotic stresses [69]. The bacteria genera *Nitrososphaera* and *Nitrospira*, as well as the species *Nitrosovibrio tenuis*, are involved in the nitrification process, while *Pseudomonas corrugata* and *Denitrobacter* spp. are involved in the denitrification process [57,65,66,70,71]. *Glomus* sp., *Glomus mosseae,* and *Glomus clarum* promote olive growing traits, photosynthesis, mineral nutrition, defense systems, and water stress tolerance [20]. The high abundance of these beneficial microbes in olive orchards was pinpointed as one of the main reasons for olive’s resilience and tolerance to several biotic and abiotic stresses (Figure 1) [62].

**Table 1 plants-13-01848-t001:** Bacterial, AMF, and fungal communities detected in the olive root endosphere and rhizosphere.

Country	Cultivar	Bacteria	Fungi	Reference
Portugal	-	Phyla: *Proteobacteria*, *Actinobacteria*, *Fermicutes*, and *Bacteroidetes*	Phyla: *Ascomycota*, *Mucoromycota*, and *Basidiomycota*	[63]
		Genera: *Pseudomonas*, *Janthinobacterium*, *Serratia*, *Rahnella*, *Pantoea*, *Micrococcus*, *Flavobacterium*, *Erwinia*, *Burkholderiales*, *Burkholderiaceae*, *Bacillus*, *Achromobacter*, and *Rhodococcus*	Genera: *Penicillium*, *Mortierellaceae*, *Pseudogymnoascus*, *Solicoccozyma*, *Umbelopsis*, *Thrichoderma*, *Tremellomyces*, *Purpureocillium*, *Papiliotrema*, *Oidiodendro*, *Niessliaceae*, *Naganishia*, *Fusarium*, *Filobasidium*, *Didymella*, *Cystofilobasidium*, *Clonostachys*, *Cladosporium*, *Botrytis*, *Boeremia*, and *Aureobasidium*	
Spain		Phyla: *Actinobacteria*, *Acidobacteria*, *Fermicutes*, *Proteobacteria*, *Gemmatimonadota*, *Bacteroidetes*, *Verrucomicrobia*, *Chloroflexi*, and *Nitrospira*	Phyla: *Ascomycota*, *Glomeromycota*, *Basidiomycota*, *Mortierellomycota*, and *Cytridiomycota*	[66]
	Picual	Genera: *Actinophytocola*, *Streptomyces*, *GP4*, *GP6*, *GP7*, *GP16*, *Ohtaekwangia*, *Sphingomones*, *Kibdelosporangium*, *Promicromonospora*, *Lentzea*, *Pseudomonas*, *Mycobacterium*, *Actinomadura*, *Neorhizobium*, *Actinoplanes*, *Rhodomicrobium*, *Bradyrhizobium*, *Nocardioides*, *Nonomuraea*, *Steroidobacter*, *Phytohabitans*, *Gematimonas*, *Blastococcus*, *Stenotrophobacter*, *Agromyces*, *Arenimicrobium*, *Gemmatirosa*, *Flavitalea*, *Solirubrobacter*, *Gaiella*, *Opitutus*, *Pseudoarthrobacter*, *Rudobacter*, *Longimicrobium*, and *Pseudonocardia*	Genera: *Diaporthe*, *Campanella*, *Mycenella*, *Panaeolus*, *Helminthosporium*, *Solicoccozyma*, *Mortierella*, *Lophotrichus*, *Cladophialophora*, *Hemycena*, *Panaeolus*, *Mycena*, *Scutellinia*, *Rhizophagus*, *Fusarium*, *Alternaria*, *Dominika*, *Aureobasidium*, *Preussia*, *Fusidium*, *Phaeomoniella*, *Hydropus*, *Dactylonectria*, *Corticum*, *Coniosporium*, *Paratricharina*, *Septoglomus*, *Botryotrichum*, *Kamienskia*, *Agaricus*, *Chrysosporium*, *Wallemia*, *Montagnula*, *Entoloma*, *Cephaliophora*, *Glomus*, and *Tricharina*	
Greece	Koroneiki and Chondrolia Chalkidikis	Phyla: *Nitrososphaerota*, *Actinobacteria*, *Acidobacteria*, *Proteobacteria*, *Bacteroidetes*, *Myxococcota*, *Verrucomicrobia*, *Gemmatimonadetes*, and *Firmicutes*	Phyla: *Ascomycota*, *Basidiomycota* and *Glomeromycota*	[65]
		Genera: *Nitrososphaera*	Family: *Glomeraceae*, *Diversisporaceae*, *Paraglomeraceae* and *Gigasporaceae*	
Tunisia	Chemlali	Phyla: Proteobacteria, Planctomycetota, and Actinobacteria Class: *Acidobacteriia*, *Actinobacteria*, *Alphaproteobacteria*, *Bacilli*, *Bacteroidea*, *Blastocatellia*, *Chororoflexia*, *Deltaproteobacteria*, *Gammaproteobacteria*, *Gemmatimonadetes*, *Phycisphaerae*, *Rubrobacteria*, *Saccharimonadia*, *Thermoleophilia*, and *Verrucomicrobiae*		[69]
Italy	Leccino	Phyla: *Proteobacteria*, *Actinobacteria*, *Bacteroidota*, *Acidobacteria*, *Myxococcota*, and *Verrucomicrobia*	Phyla: *Ascomycota*, *Basidiomycota* and *Glomeromycota*	[59]
			Genera: *Glomus*, *Dominikia*, and *Rhizophagus*	
Italy	Frantoio		Phyla: *Glomeromycota* Family: *Glomeraceae*	[72]
			Genera: *Glomus*	
Spain	36 ^1^	Phyla: *Actinobacteria*, *Acidobacteria*, *Proteobacteria* (classes *Alphaproteobacteria*, *Gammaproteobacteria* and *Deltaproteobacteria*), *Gemmatimonadota*, *Fermicutes*, and *Bacteroidetes*	Phyla: *Ascomycota*, and *Basidiomycota* Class: *Sordariomycetes*, *Eurotiomycetes*, *Agaricomycetes*, *Dothideomycetes*, *Orbiliomycetes*, *Pezizomycetes*, and *Leotiomycetes*	[62]
		Genera: *Actinophytocola*, *Streptomyces*, *Pseudonocardia*, *Pseudomonas*, *Gp6*, *Gp4*, *Gp7*, *Rhizobium*, *Sphingomonas*, *Gemmatimonas*, *Bacillus*, *Bradyrhizobium*, *Ensifer*, *Rubrobacter*, *Steroidobacter*, *Saccharothrix*, *Ohtaekwangia*, *Mycobacterium*, and *Nonomuraea*	Genera: *Canalisporium*, *Macrophomina*, *Malassezia*, *Minimelanolocus*, and *Aspergillus*	
Italy	Maiatica	Phyla: *Proteobacteria*, *Actinobacteria*, *Cyanobacteria*, *Nitrospirota*, and *Nitrososphaerota*		[71]
		Genera: *Roseomonas*, *Microcoleus*, *Pseudomonas*, *Mesorhizobium*, *Bradyrhizobium*, *Sinorhizobium*, *Denitrobacter*, *Acidisphaera*, *Phaeospirillum*, *Azospirillum*, *Frankia*, *Nostoc*, *Acaryochloris*, *Nitrospira*, *Nitrosovibrio*, and *Nitrososphaera*		
Italy	Maiatica	Phyla: *Actinobacteria*, *Proteobacteria*, *Fermicutes*, *Nitrospira*, *Acidobacteria*, *Chloroflexi*, *Bacteroidetes*, and *Verrucomicrobia*		[57]
		Genera: *Rhodanobacter*, *Ramlibacter*, *Pigmentiphaga*, *Roseomonas*, *Rubellimicrobium*, *Ancylobacter*, *Rhodoplanes*, *Sporosarcina*, *Caldilinea*, *Adhaeribacter*, *Flavisolibacter*, *Solirubrobacter*, *Conexibacter*, *Rubrobacter*, *Microlunatus*, *Mycobacterium*, *Arthrobacter*, and *Cellulomonas*		
Spain	Lechín, Manzanillo, Nevadillo, and Picual	Phyla: *Proteobacteria* and *Nitrospirota* Species: *Nitrosospira tenuis* and *Nitrospira* sp.		[70]
Portugal	Cobrançosa		Phyla: *Ascomycota* Species: *Fusarium oxysporum*, *Phomopsis columnaris*, *Macrophomina phaseolina*, *Trichoderma* spp.1, *Trichoderma gamsii*, *Lophiostoma corticola*, *Paecilomyces verrucosus*, and *Penicillium restrictum*	[58]
Spain	-		Phyla: *Glomeromycota* Genera: *Glomus*, *Archaeospora*, *Diversispora*, *Claroideoglomus*, and *Paraglomus*	[73]
Spain	Wild olives	Phyla: *Fermicutes*, *Proteobacteria*, *Bacteroides*, and *Actinobacteria* Genera: *Pseudomonas*, *Bacillus*, *Paenibacillus*, *Actinobacter*, *Rahnella*, *Erwinia*, *Rhodococcus*, and *Chryseobacterium*		[64]


^1^ Klon-14-1812, Chemlal de Kabylie, Kalamon, Koroneiki, Mastoidis, Mavreya, Megaritiki, Shengeh, Myrtolia, Uslu, Dokkar, Majhol, Jabali, Maarri, Barri, Abou Kanani, Abou Satl Mohazam, Verdial de Vélez, Abbadi Abou Gabra, Piñonera, Temprano, Picual, Picudo, Menya, Morrut, Manzanillera de Huercal Overa, Manzanillera de Sevilla, Forastera de Tortosa, Arbequina, Llumeta, Leccino, Frantoio, Barnea, Grappolo, Mari.

**Table 2 plants-13-01848-t002:** Beneficial effects of plant growth-promoting bacteria (PGPB) and fungi (PGPF) on olive trees.

**PGPB**	**Benefits**	**References**
Classes *Actinobacteria*, *Thermoleophilia and Bacilli*	Adaptation to environmental stresses, e.g., arid and dry conditions	[69]
Phyla *Acidobacteria* (classes *Acidimicrobiia*, *Blastocatellia* and *Rubrobacteria*), *Planctomycetota* (class *Phycisphaerae*), *Verrucomicrobia* (class *Verrucomicrobiae*), *Proteobacteria* (class *Gammaproteobacteria*) and *Gemmatimonadetes*	Protection against biotic and abiotic stresses	[69]
Genera *Pseudomonas*, *Bacillus* and *Streptomyces*	Biological control agents	[57,62,64]
Phyla *Actinobacteria* and *Proteobacteria* (*Curtobacterium flaccumfaciens and Methylobacterium mesophilicu*	Biological control agents	[57]
Phylum *Actinobacteria* (e.g., *Streptomyces* spp.)	Production of secondary metabolites that help in pathogen defense	[66,67]
Phylum *Firmicutes*	Antagonists and biological control agents	[57]
**PGPF**	**Benefits**	**References**
Genus *Aureobasidium*	Suppress anthracnose	[65,66]
Genus *Cladophialophora*	Biological control agents	[66]
Genera *Preussia* and *Diaporthe*	Produce secondary metabolites that help to control the development of some plant pathogens	[20,66]

Many AMF are known to host intracellular bacteria, such as endobacteria, which live within fungal hyphae and spores, and can affect host function and subsequent plant–fungus interactions [59,74]. AMF endobacteria may improve fungal ecological fitness and promote antioxidative responses in both fungal and plant hosts, but many (e.g., *Burkholderia rhizoxinica*) can also induce a pathogenicity factor in the host, enabling the host fungus to cause disease in the plant [74]. Studies related to the role of AMF endophytic bacteria and their functional significance are still very limited in terms of olive. Chialva et al. [59] that the different patterns of abundance of AMF and their endobacteria can reveal the presence of interactions in olive roots, but the significance of these interactions in conferring some relevant traits to AMF and olive, for example resilience to (a)biotic stress, deserves further studies.

### 3.2. Influence of Biotic and Abiotic Factors on Microbiota Communities

The Mediterranean area accounts for nearly 80% of the olive cultivation area worldwide, with the world’s largest olive oil and table olive producer being Spain, followed by Italy, Tunisia, Turkey, and Greece [7]. For comprehensive knowledge on olive belowground microbiota, a wide range of regions of olive cultivation must be surveyed. However, most of the studies available characterize the microbes in the rhizosphere and root endosphere in the most representative European olive countries, Spain, Italy, Portugal, and Greece (Table 1). Microbiota communities in other important regions of olive production, like Tunisia and Turkey, or in new cultivation regions, like Brazil and China (Table 1), with different climatic conditions and soil characteristics, remain poorly explored. Moreover, the use of similar methods and bioinformatic analysis is essential to maximize the consistency of results and enable data comparison. 

Abiotic and biotic factors can shape the composition and diversity of olive microbiota [24]. The information available on abiotic stress effects on the olive rhizosphere and root endosphere microbiota focuses mostly on the impact of agronomic practices and environmental conditions [24]. Regarding the biotic factors, the influence of pathogens, cultivar/genotype, and plant phenological stage has also been analyzed [24]. The shifts in olive microbiota caused by pathogens, like *Xylella fastidiosa*, *Verticillium dahliae* and *Colletotrichum* spp., have been extensively described and will not be addressed in this review.

The effects of tillage practices on rhizosphere fungal and bacterial communities in a Portuguese olive orchard (Mirandela, Bragança district) were studied by Kacem [63]. Tilled vs. untilled soils influenced fungal and bacterial community profiles, with tillage reducing fungal abundance. The genera *Mortierellaceae* and *Janthinobacterium* were present exclusively in tilled soils, while the genera *Pseudogymnoascus*, *Solicoccozyma*, and *Rhodococcus* were only found in untilled soils. In an olive orchard (cultivar Maiatica) situated in Southern Italy and subjected to two agricultural management practices, sustainable and conventional, the sustainable system presented a higher number and diversity of microorganisms [57]. For instance, bacteria from the genus *Rhodanobacter* (which includes many microorganisms involved in acidic denitrification in soils) were found exclusively in soils of the sustainable system, and was a high abundance of bacteria of the order *Rhizobiales* (N-fixing bacteria). The high abundance of N-fixing bacteria in the sustainable system’s soil was associated with the presence of the dense root systems of cover crops, like spontaneous Fabaceae, whose root nodules host symbiont N-fixing bacteria, acting indirectly by providing higher soil nitrogen availability for olive trees. The characterization of soil microbial communities from another Italian olive orchard (cultivar Maiatica), subjected to sustainable and conventional management practices, showed that the bacterial community diversity profiles (bacterial taxa related to nitrogen transformation) of the two systems were similar [71]. However, the abundance of the bacteria varied. In the conventional system, the genera of the free N-fixing bacteria *Roseomonas* and *Microcoleus* and the denitrifying bacteria *Pseudomonas* were the most abundant. In turn, the bacterial genera most abundant in the sustainable system were the symbiotic N-fixing *Mesorhizobium* and *Bradyrhizobium* genera, and nitrifying bacteria *Nitrospira* spp., and *Nitrosovibrio* and *Nitrososphaera*. Palla et al. [72] analyzed the diversity and composition of native AMF in the roots of olive trees (cultivar Frantoio) from an Italian experimental farm (high-density) at the University of Pisa and subjected them to different management treatments (shallow tillage and permanent green cover). The management practices shaped the AMF community composition of olive roots. The predominant phylum was *Glomeromycota* (family *Glomeraceae*), with the species *Glomus* spp. being found in the highest amounts in soils from the green cover treatment. In a Spanish orchard (Lupión, Jaén, Spain), Wentzien et al. [66] evaluated the influence of organic (sheep manure) and conventional management practices in the rhizosphere and root endosphere microbial communities of the olive cultivar Picual. The manure amendment enhanced the richness and diversity of the microbial community and was beneficial for soil nutrient content. In terms of the bacterial taxonomical profile, both systems were mostly dominated by the phyla *Actinobacteria*, *Acidobacteria*, and *Proteobacteria*. At the genus level, in the organic system, *Actinophytocola*, *Streptomyces*, and *Kibdelosporagium* were the most abundant in the root endosphere, while *GP6*, *Ohtaekwangia*, and *Sphingomones* dominated the rizosphere. In the conventional system, the root endosphere was dominated by the genera *Kibdelosporagium* and *Pseudonocardia*, while in the rizosphere, the genera *GP6*, *Ohtaekwangia*, and *GP4* were dominant. Regarding the fungal rhizosphere community, the phyla *Ascomycota*, *Glomeromycota*, and *Basidiomycota* were the most abundant in both the root endosphere and rizosphere in both conventional and organic systems. Additionally, in the organic system, the phylum *Mortierellomycota* was also prevalent. At the genus level, in the organic system, the most abundant genera in the root endosphere were *Diaporthe*, *Campanella*, *Mycenella*, *Panaeolus*, and *Helminthosporium*, and in the rizosphere, the most abundant genera were *Solococcozyma*, *Mortierella*, and *Lophotrichus*. In the conventional system, the most abundant genera were *Cladophialophora*, *Campanella*, and *Hemycena* in the root endosphere, while in the rhizosphere, the most abundant genera were *Solococcozyma* and *Tricharina*. In a study conducted in Spain on several olive orchards (Jaén, Granada, Córdoba, and Sevilla areas) with the cultivars Lechín, Manzanillo, Nevadillo, and Picual, exposed to different agricultural practices (conventional vs. organic agriculture), climatic conditions, and soil properties, the bacterial communities in the rhizosphere were mostly dominated by the *Nitrosospira tenuis* and *Nitrospira* spp. [70]. *Nitrosospira tenuis* was less abundant in the cultivar Picual, and in the rhizosphere of the conventionally managed orchards. 

A study conducted in Tunisia by Marasco et al. [69], on olive trees (cultivar Chemlali) selected from five locations with different climatic conditions (Zaghouan: higher-semiarid; Chraitia and Gafsa: higher-arid; Matmata: middle-arid; and Neffatia: lower-arid), demonstrated that the main abundant classes of bacteria in root tissues were *Actinobacteria*, *Alphaproteobacteria*, and *Gammaproteobacteria*, and those in the rhizosphere and bulk soil were *Actinobacteria*, *Acidobacteria*, *Alphaproteobacteria*, *Gammaproteobacteria*, *Bacteroidia*, *Thermoleophilia*, and *Sacharimonadia*. Aridity particularly affected bacterial communities associated with the root systems and bulk soils. The abundance of bacteria from the phyla *Actinobacteria* (*Actinobacteria* and *Thermoleophilia*), *Chloroflexi*, and *Firmicutes* (*Bacilli*) increased with aridity, while that of *Acidobacteria* (*Acidimicrobiia*, *unclassified Acidobacteria* and *Blastocatellia*), *Planctomycetes* (*Phycisphaerae*), *Verrucomicrobiae*, *Gammaproteobacteria*, *Rubrobacteria*, and *Gemmatimonadetes* decreased. Chialva et al. [59] characterized the fungal and bacterial profile of a 20 year-old Italian orchard (northern Italy, Pinerolo). Root sample analyses from the cultivar Leccino, a cold-hardy cultivar, and Frantoio, a cold-susceptible one, were conducted in two seasons: spring and winter. Bacteria from the phylum *Proteobacteria* were the most abundant component in these soils, followed by the phyla *Actinobacteriota*, *Bacteroidota*, *Myxococcota*, *Acidobacteriota*, and *Verrucomicrobiota*. Regarding AMF communities in soils, fungi from the *Glomeromycota* phylum, in particular the *Glomeraceae* and *Claroideoglomeraceae* families, were present in higher percentages. The most representative genera were *Glomus*, *Dominikia*, and *Rhizophagus*. Neither the season nor the cultivar seemed to have influenced the qualitative microbial profile, but both factors affected microbial abundance, particularly in the cultivar Leccino. Martins et al. [58] characterized the fungal microbial community in root samples from different seasons and in three districts of Portugal (Bragança, Mirandela and Carrazeda de Ansiães). Several endophytic fungi were identified, with *Trichoderma* spp., *Phomopsis columnaris*, and *Fusarium oxysporum* being the most prevalent. *Trichoderma* spp. and *F. oxysporum* were more abundant in November, while *P. columnaris* was more abundant in the three regions surveyed, and *F. oxysporum* was more common in Carrazeda de Ansiães.

Fernández-Gonzáles et al. [62] described microbiota associated with the rhizosphere and olive roots of a collection of 36 cultivars located in the World Olive Germplasm Collection in Córdoba (Spain). The bacterial genera *Actinophytocola*, *Streptomyces*, *Pseudonocardia*, *Gp6*, *Gp4*, *Rhizobium*, *Sphingomonas*, and *Gemmatimonas* and the fungal genera *Canalisporium*, *Macrophomina*, and *Aspergillus* were the most abundant in the olive root endosphere and rhizosphere. Microbial communities were influenced by the factor cultivar, which particularly affected the bacterial community. The phylum *Actinobacteria* was the most dominant endophytic community in the cultivars Chemlal de Kabylie, Llumeta, and Mavreya. The genus *Flavitalea* was more abundant in roots of the cultivar Myrtolia and absent in Uslu. In the rhizosphere, the genera *Gp6*, *Gp4*, and *Gp7* (*Acidobacteria* phylum) were the most prevalent in the cultivars Temprano and Barnea. In olive orchards in Southern and Northern Greece, Kakagianni et al. [65] determined the prokaryotic, fungal, and AMF microbiota profile in the olive roots and rhizosphere of two cultivars, Koroneiki and Chondrolia Chalkidikis, during different vegetative stages. In both cultivars, similar prokaryotic, AMF, and fungal communities were found but in different abundances. The prokaryotic and fungal communities’ patterns varied within the vegetative stages. The roots and the soil were dominated by the bacteria classes *Actinobacteria*, *Thermoleophilia*, *Alphaproteobacteria*, *Gammaproteobacteria*, *Nitrososphaeria*, and *Bacilli*. *Nitrososphaeria* (soil ammonia-oxidizing bacteria) was more abundant in the rhizospheric soil and Chondrolia roots. Regarding the fungal communities, soil and roots presented a high abundance of the order *Hypocreales*, *Pleosporales*, and *Arganicales*. Concerning the AMF families in the soil, Chondrolia roots were dominated by the order Diversisporaceae (phylum *Glomeromycota*), while Koroneiki was dominated by the family *Glomeraceae*.

## 4. The Role of Microorganisms in Alleviating Abiotic Stress’s Negative Effects on Olive Trees

The Mediterranean area is facing frequent and more intense drought periods, heat waves, and soil salinization. An orchard’s rhizosphere can harbor microorganisms with the potential to improve plants’ resilience and performance under abiotic stress [75]. In fact, the adaptability of olive trees to a high diversity of environmental conditions has been attributed to the native fungi and bacteria that naturally colonize it [76]. Various studies have explored the use of fungal or bacterial inocula to improve olive performance under several environmental conditions. However, most of these experiments were conducted using young olive plants, mostly growing in pots or under nursery conditions, with fewer studies focusing on mature olive trees under field conditions. 

### 4.1. Plant Growth-Promoting Fungi

The potential benefits of AMF communities isolated from olive orchards, or of other species, as well as those present in commercial mixtures, have been tested in olive plants under different stress conditions. 

For instance, Calvo-Polanco et al. [77] assessed the effects of different AMF populations isolated from soils of semi-arid and humid olive orchards on olive performance under drought conditions. Potted young olive plants (cultivar Picual, moderately tolerant to drought) inoculated with AMF populations from a humid region (mainly composed of *Glomeromycota* spp., *Glomus* spp., and *Funneliformis* spp.) showed improved performance under well-watered and water deficit stress conditions. This inoculum seems to modulate the expression of aquaporin genes, phosphorylation, and root water transport. The use of a commercial inoculum during olive plant production has also proven to be very effective in reducing stresses related to field conditions. In a study conducted in Morocco, the root system of one-year-old potted olive trees (cultivar Picholine marocaine) was inoculated with an AMF (*Rhizophagus irregularis*) and subjected to deficit irrigation [78]. AMF inoculation improved olive growth indices, water status, and performance (showing increased levels of proline, sugar, and δ^13^C) under water deficit conditions. However, plant performance did not reach the values obtained under well-watered conditions. Inoculation of olive plantlets (growing in pots) with spores of *Rhizophagus irregularis* DAOM 197198 and *Glomus intraradices* DAOM 197198 improved plant growth (shoot biomass production) under water deficit conditions. AMF inoculation prevented the decline in the plant water potential to critical values, induced the accumulation of sugars, sugar alcohols (mannitol), and antioxidants (catechin, luteolin 7-*O*-glucoside, and oleuropein) in roots, and modulated the phytohormone profile, which seems to contribute to the protection of olive plants against stress [79]. Commercial AMF mixtures were tested for their potential to improve olive performance during transplanting and establishment in field. Lopes et al. [80] tested the effect of two commercial biostimulants, one composed of a mixture of five AMF (*Rhizophagus irregularis*, *Funneliformis mosseae*, *F. geosporum*, *F. coronatum*, and *Claroideoglomus claroideum*) and the second containing propagules of nine species of ectomycorrhizal fungi (*Pisolithus tinctorius*, four species of the genus *Rhizopogon*, two species of the genus *Scleroderma*, and two species of the genus *Laccaria*) and nine AMF (seven species of the genera *Glomus*, *Rhizophagus irregularis*, and *Paraglomus brasilianum*). The experiment was conducted on young olive trees (~60 cm in height) transplanted in the field under rainfed conditions (Mirandela, Portugal). AMF inoculation increased the growth of young trees, and alleviated drought field stress and Ca and Mg disorders [80]. Another commercial product based on *Trichoderma harzianum* was tested on young olive potted plants exposed to water deficit conditions [81]. This PGPF improved the total phenol contents and antioxidant activity of olive under water deficit conditions. Also, Ouledali et al. [82] assessed the effect of AMF inoculation on one-year-old olive plants under dry-field conditions using a commercial inoculum containing six *Glomus* species (*G. etunicatum*, *G. microaggregatum*, *G. intraradices*, *G. claroideum*, *G. mosseae*, and *G. geosporum*). Inoculated olive plants showed higher turgor potential, better stomatal response, better osmotic adjustment, and improved mineral uptake. Moreover, AMF inoculation improved recovery from drought stress [82]. The inoculation of young potted olive plantlets (cultivar Chétoui) with the AMF *Rhizophagus irregularis* improved the photosynthesis, stomatal conductance, water status, monosaccharide and disaccharide content, mannitol content, and content of phenolic compound (e.g., oleuropein, tyrosol, luteolin 7-*O*-glucoside, apigenin 7-*O*-glucoside, and verbascoside) under reduced irrigation, leading to an enhanced capacity to avoid drought damages [83]. Similarly, Bompadre et al. [84] inoculated olive cuttings (cultivar Manzanillo) with two strains of *Rhizophagus irregularis*. The combination of the two AMF strains promoted olive growth in the nursery and also activated plant antioxidant defenses (e.g., catalase and superoxide dismutase).

Some AMF can enhance the tolerance of olive plants to salinity stress. This is particularly significant given that the reuse of treated wastewater rich in salts is a common practice in some Mediterranean agricultural regions experiencing water scarcity. For instance, inoculation of young potted olive plants (cultivar Chetoui) with *Glomus deserticola*, *Gigaspora margarita*, and a mixture of *G. deserticola* and *G. margarita* alleviated the negative effect of irrigation with salt-enriched wastewater [85]. AMF increased plant water status, total fresh and dry weights, and photosynthetic activity, and also reduced Na^+^ and Cl^−^ accumulation in leaves and roots [85]. These AFM also led to a higher concentration of proline, soluble sugars, and macro- and micro-nutrients, as well as to a better antioxidant defense system compared with that of non-mycorrhizal plants [85,86]. Nevertheless, inoculation of olive cuttings (cultivar Koroneiki) with natural fungal communities (AMF and endophytic fungi) collected in six different extreme Mediterranean sand dunes induced different plant responses to salt stress (50 mM and 100 mM NaCl) [86]. Some inocula had a positive effect on plant growth and photosynthesis, while others did not improve plant traits, possibly due to high salinity sensitivity [86]. These authors suggested that the different osmotic responses of fungal communities may be related to adaptative traits, so some inocula could be undesirable if applied to allochthonous agronomic plants [86]. 

### 4.2. Plant Growth-Promoting Bacteria 

Overall, PGPB are indicated to play an essential role in boosting plant growth, health, and yield. In fact, they can be a valuable source of biofertilizers, biostimulants, and biocontrol agents; therefore, they must be incorporated into organic olive production [87]. The potential for several bacteria to improve olive performance under stress conditions, such as saline soils, calcareous soils, soils with an excess of heavy metals or under drought conditions, have been investigated. Some studies used bacteria inocula isolated from the soil or rhizosphere of olive or other species [88,89,90], while others opted to use material from PGPB collections [91,92]. Recently, Dias et al. [91] demonstrated that in young potted olives, inoculation of the rhizosphere with *Pseudomonas reactans* promoted plant performance under well-irrigated and irrigation deficit conditions. Under well-watered conditions, *P. reactans* increased leaf dry biomass and soil C and N availability, and under water deficit stress conditions, it promoted leaf water status, N content, photosynthesis, carbohydrates, and total antioxidant capacity, and reduced oxidative stress. 

In turn, Azri et al. [89] used PGPB consortia as an inoculum, obtained from bulk and rhizosphere soil collected from olive orchards (Tunisia) of the cultivar Chetoui (a drought-sensitive olive cultivar). Potted olive plantlets from the drought-sensitive cultivar Chetoui and from the drought-tolerant cultivar Chemlali were inoculated with these PGPB consortia, and after two years, water deficit stress was observed. The PGPB consortia induced stress tolerance in both olive cultivars, despite some differences in the metabolite response. Chetoui plants inoculated with the PGPR consortia and exposed to stress produced more D-mannitol, allo-inositol, galactinol, soluble sugars (glucose and sucrose), squalene, quinic acid, and tocopherol in roots than Chemlali plants did in their roots. Similarly, Galicia-Campos et al. [92] inoculated young olive plants (cultivars Arbosana and Arbequina) potted (with soil from marshes of Guadalquivir river, Spain) with 10 PGPB (G7, H47, K8, L24, L36, L44, L56, L62, L79, and L81) isolated from the rhizosphere of *Pinus pinea* and *Pinus pinaster* from Huelva (Spain). The PGPB improved plant fitness, particularly due to the increase in photosynthetic pigments and antioxidant pools (ascorbate and glutathione). Galicia-Campos et al. [92] also inoculated young potted olive plants with *Bacillus simplex* (G7:OP324816) and observed that plants treated with these PGPB and exposed to saline conditions showed improved stress tolerance. *Bacillus simplex* increased photosynthesis (photosynthetic efficiency and net CO_2_ assimilation rate), water use efficiency, energy dissipation mechanisms, and antioxidants (total phenols, flavonoids, and proline), and promoted osmoregulation, particularly in salt-stressed olive plants. Moreover, *B. simplex* activated the abscisic acid-mediated pathway (upregulating the genes related to abscisic acid biosynthesis and downregulating the repression of phosphatases of the PLC2 family) and the genes related to ion homeostasis that help olive plants to cope with salt stress. 

PGPR can also help mitigate other abiotic stresses like high soil calcicity, a common issue in the Mediterranean region, and they also function effectively across a broad range of temperatures and soil pH levels. Maksoud et al. [93] found that the yield and production of high-quality fruits of olive trees grown in calcareous soil were increased after applying the commercial biofertilizer ‘Phosphorine’, which contains an active strain of phosphate-soluble bacteria (PSB) in combination with an antioxidant (ascorbic acid or citric acid). In another study, when an *Azotobacter chroococcum* bacterial suspension was combined with aqueous extracts from three types of compost—farmyard manure, town refuse, and sewage sludge—to treat olive trees (Chemlali cultivar) grown in calcareous soil, the significant positive effects on vegetative growth, yield, and flesh oil content, particularly at a 30% compost rate, were further enhanced [94]. Regarding pH and temperature variation, it was shown that the bacterial strains *Azospirillum brasilense* BR11001t, *Azospirillum amazonense* BR11040t, *Herbaspirillum seropedicae* BR11175t, and *Burkholderia brasilensis* BR11340t promoted root induction across a broad spectrum of pH levels (5–9) and temperatures (15–35 °C) [95]. The combined inoculation of AMF and PGPR strains, applied either alone or together with other constituents, such as antioxidants or humid acids, was also investigated [20]. El-Shazly and Ghieth [96] demonstrated that the combination of PGPR (*A. chroococcum*) and AMF (*Glomus macrocarbium*) with the application of humic acid mitigated the adverse effects of salinity, resulting in higher total microbial counts, increased bacterial densities, greater AMF infection rates, and enhanced soil enzymatic activity. The study attributed these improvements to the production of phytohormones such as indole-3-acetic acid, gibberellins, cytokinins, and ethylene, along with nitrogen fixation, phosphate availability, and nutrient solubilization.

## 5. Conclusions

Olive is one of the most important crops in the Mediterranean area, and the demand for olive products has increased substantially in the last decade, particularly due to its recognized beneficial effects on human health. Despite the high adaptability of olive trees to harsh environmental conditions, the increase in climate weather events due to climate change represents a threat to olive culture, leading to reductions in orchard productivity. In response, olive cultivation management practices have been adjusted to enhance the sustainability of this sector. The potential of some fungi and bacteria to promote plant performance has long been recognized, and their use has been recognized as a worthy strategy to promote the sustainability of agricultural systems. Therefore, the implementation of sustainable olive agricultural practices can rely on the use of beneficial microorganisms to promote productivity and stress tolerance. Knowledge on olive belowground microbiota, as well as knowledge on the influence of biotic and abiotic factors on these communities, is crucial for selecting specific fungi and bacteria that enhance olive traits under specific conditions. Several fungi and bacteria communities were identified mostly in orchards of the most representative EU olive producers, where the belowground microbiota richness is mostly represented by the bacterial phyla *Actinobacteria*, *Proteobacteria*, *Acidobacteria*, *Fermicutes*, *Bacteroidetes*, and *Verrucomicrobia*, and by the fungal communities belonging to the phyla *Ascomycota*, *Basidiomycota*, and *Glomeromycota*. Some of these microorganisms (e.g., the genera *Pseudomonas*, *Fusarium*, *Bacillus*, *Streptomyces*, *Cladophialophora*, *Aureobasidium*, *Diaporthe*, *Preussia*, *Nitrososphaera*, *Nitrospira*, *Denitrobacter*, and *Glomus*) have been shown to be beneficial, promoting the physiological performance of several species, including olive, under different environmental conditions. However, abiotic factors can shape microbiota diversity and composition in the olive rhizosphere and roots. Additionally, management practices, like tillage and type of fertilization, season, aridity, geographic location, and cultivar, can also affect the qualitative and quantitative microbiota profile of belowground olive orchards. The use of microorganisms isolated from olive orchards or from other species has been proven to be effective in enhancing olive growth and resilience. However, more research on the role of microorganisms in reducing drought, salinity, and other stresses, like heat, on old/mature olive trees in field orchards should be conducted. The importance of AMF endophytic bacteria and their interaction with fungi and the host plant (olive) should be further explored. Also, the development of guidelines or/and protocols for methods and data analysis is essential to improve knowledge on microbial communities and maximize the consistency of results. Expanding research on the olive orchard microbiota to other regions with high olive production outside the EU and to new regions of olive cultivation, like Brazil, Argentina, and China, is also necessary since the high diversity of climatic conditions and soil characteristics of these areas can influence bacterial and fungal communities’ assembly. Thus, knowledge on microbial composition and on the diversity of belowground olive orchards remains fragmented. Since the desired production of high-quality olive oil in high quantities relies on numerous factors, biotic and abiotic, which act in an integrated manner and influence each other, field studies evaluating olive tree cultivars, along with the physical, chemical, and biological characteristics of the soil and the climate, are of the utmost importance. However, the strategies presented here may contribute to the design of more sustainable agronomic practices and help to develop new approaches to face climate environmental challenges and to meet the growing demand of olive products. 

## Figures and Tables

**Figure 1 plants-13-01848-f001:**
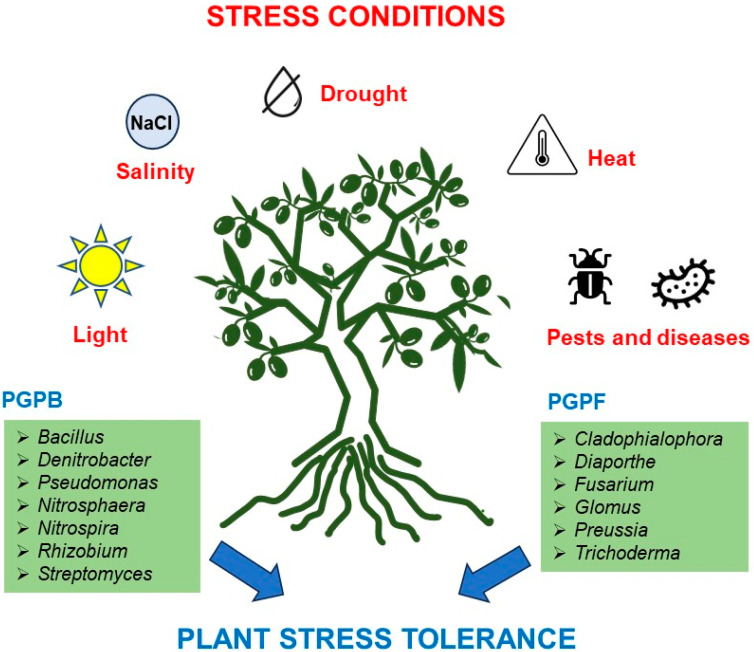
Examples of some genera of plant growth-promoting bacteria and fungi that help olive plants to cope under stress conditions.

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
