# Peer review of "Olive Tree Belowground Microbiota: Plant Growth-Promoting Bacteria and Fungi"

_plants, 2024, doi:10.3390/plants13131848_

Round 1

Reviewer 1 Report

Comments and Suggestions for Authors

Comments to the Author:

I would like to extend my congratulations to the authors on their article, " Olive tree belowground microbiome: plant growth-promoting bacteria and fungi". I acknowledge the importance to collect and update the data pertaining to the olive tree microbiota and its structure influenced by different host and environmental factors. However, I have provided some suggestions aimed at enhancing the presentation of the review.

Specific comments to the authors:

I found some typos, please focus on them and on major comments:

- throughout the text, replace "genus” with “genera” when more of one are listed (for example in lines 62, 81, 228, 232, 236, 239 and so on) and specify the taxon before a list (for example lines 70, 72, 83, 238, etc.)

- line 91: "In fact” is adversative, it is better to use “Indeed”

- lines 99, 104, 124: please replace "region" with “regions” and modify the verbs accordingly

- paragraph 3.1: in my opinion, for completeness of the discussion it would be appropriate to also report some information on endohyphal bacteria (AMF endobacteria) and their potential effects on fungal and plant hosts. Furthermore, this topic should also be reported as a knowledge gap that should be explored further.

- from line 209 to 212: the reference [58] is missing and never cited in paper, please correct bibliography

- line 269: in table 1 it would be better to add a column reporting the PGP properties of microbial genera covered in the text (or prepare another Table)

- paragraph 3.2: should be reorganized to ensure a linear reading flow by moving the paragraphs. Starting from line 293, the impact of agronomic practices and then that of environmental conditions should be reported in order as announced in line 287. Fox example the studies of Kacem and Wentzien et al. should be stated together. As well as Martins et al. and Chialva et al. on seasonality.

- line 448: “(cultivar Chétoui))” please remove one round bracket

- line 519: “(Chemlali. cultivar)” please remove full stop

Sincerely

Author Response

Dear Editor,We would like to acknowledge the valuable comments, remarks and suggestions made by the reviewers that helped us to improve the quality of the manuscript. We marked in red colour the changes made in the manuscript suggested by the Reviewers. Moreover, we answer the Reviewers’ comments.

Reviewer 1

Comments to the Author:

I would like to extend my congratulations to the authors on their article, " Olive tree belowground microbiome: plant growth-promoting bacteria and fungi". I acknowledge the importance to collect and update the data pertaining to the olive tree microbiota and its structure influenced by different host and environmental factors. However, I have provided some suggestions aimed at enhancing the presentation of the review.

Specific comments to the authors:

I found some typos, please focus on them and on major comments:

Comment: - throughout the text, replace "genus” with “genera” when more of one are listed (for example in lines 62, 81, 228, 232, 236, 239 and so on) and specify the taxon before a list (for example lines 70, 72, 83, 238, etc.)

Response: We replaced "genus” by “genera” when applicable and introduced the taxon when not described. Please see the new version of the document. 

Comment: - line 91: "In fact” is adversative, it is better to use “Indeed”

Response: The term was corrected. Please see line 91 of the new version of the document.

Comment: - lines 99, 104, 124: please replace "region" with “regions” and modify the verbs accordingly

Response: We changed “region” to “regions”. Please see lines 100, 105, and 125 of the new version of the document.

Comment: - paragraph 3.1: in my opinion, for completeness of the discussion it would be appropriate to also report some information on endohyphal bacteria (AMF endobacteria) and their potential effects on fungal and plant hosts. Furthermore, this topic should also be reported as a knowledge gap that should be explored further.

Response: We agree with the Reviewer opinion. In fact, studies related to AMF endophytic bacteria are very limited. We add some information about this issue in the section 3.1 and conclusions.

Comment: - from line 209 to 212: the reference [58] is missing and never cited in paper, please correct bibliography

Response: We added the missed reference. Please see line 146 of the new version of the document.

Comment: - line 269: in table 1 it would be better to add a column reporting the PGP properties of microbial genera covered in the text (or prepare another Table)

Response: We added the required information in a new table (Table 2). Please see the new version of the manuscript.

Comment: - paragraph 3.2: should be reorganized to ensure a linear reading flow by moving the paragraphs. Starting from line 293, the impact of agronomic practices and then that of environmental conditions should be reported in order as announced in line 287. Fox example the studies of Kacem and Wentzien et al. should be stated together. As well as Martins et al. and Chialva et al. on seasonality.

Response: Previously we presented these studies accordingly to the year of publication and country where the experiment was conducted. In the new version of the manuscript, we follow the Reviewer suggestion. We started to describe the studies related to the impact of the agronomic practices, environmental conditions, the effect of cultivars and, at the end, the influence of plant phenological stage. Please see the new version of the manuscript, section 3.2.

Comments: - line 448: “(cultivar Chétoui))” please remove one round bracket.

                    - line 519: “(Chemlali. cultivar)” please remove full stop

Response: We corrected the text according to the Reviewer comment (please see lines 473 and 544 of the new version of the manuscript). 

Reviewer 2 Report

Comments and Suggestions for Authors

I am sending a few minor comments that do not affect my overall assessment but require correction or clarification.

l.2, 25 and many others – please instead microbiome use microbiota = def. all organisms in particular niche; microbiota = genomes of all organisms in particular niche; unless used in the second sense

l. 84 I would suggest to use ‘many plant species’ instead ‘several ..’ which is closer to the current

l.137, 139, 141 and 144 – [29] - it is enough for there to be one reference at the end of the paragraph in line 145

l. 156 - The biochemical concept is included in the metabolomics approach, so it can be omitted

l. 226 -in accordance with the requirements of mdpi journals, the table should appear after its first reference

l. 232 Did the authors mean pathogenic Bacillus, Pseudomonas and Fusarium in this sentence? Fusarium oxysporum, which is also mentioned below, is a generally dangerous phytopathogen as well P. syringae.

l. 240 instead ‘such as’ I would propose ‘having antimicrobial activity’

l. 266 – please improve quality of figure now it is difficult to read

l. 364, 371 - Does the presence of rhizobia have any significance for the health and growth of olive trees? I ask because these bacteria enter into symbiosis mainly with the roots of legumes and then they can provide the host with N.

page 10 - Were the described research results, conducted by different research teams on the microbiota associated with the olive growth environment, carried out with similar parameters for bioinformatic analyses? Such parameters may influence the results of the described microbial communities. If not, in my opinion it should be discussed briefly.

l. 388 - on what basis it was indicated that endophytic strains dominated in the rhizosphere. It is true that facultative endophytes can temporarily migrate between plant tissue and soil. However, this sentence is unclear to me.

l. 508 – ABA - no explanation of the abbreviation

Comments on the Quality of English Language

minor improvements are necessary

Author Response

Dear Editor,We would like to acknowledge the valuable comments, remarks and suggestions made by the reviewers that helped us to improve the quality of the manuscript. We marked in red colour the changes made in the manuscript suggested by the Reviewers. Moreover, we answer the Reviewers’ comments.

Reviewer 2

I am sending a few minor comments that do not affect my overall assessment but require correction or clarification.

Comment: l.2, 25 and many others – please instead microbiome use microbiota = def. all organisms in particular niche; microbiota = genomes of all organisms in particular niche; unless used in the second sense

Response: We unknowledge the clarification and we corrected the term. Please see the new version of the manuscript.

Comments:

  1. 84 I would suggest to use ‘many plant species’ instead ‘several ..’ which is closer to the current

 l.137, 139, 141 and 144 – [29] - it is enough for there to be one reference at the end of the paragraph in line 145

  1. 156 - The biochemical concept is included in the metabolomics approach, so it can be omitted
  2. 226 -in accordance with the requirements of mdpi journals, the table should appear after its first reference

Response: We correct the text according to the Reviewer suggestion. Please see lines: 84, 146 and 157 and table 1 in the new version of the manuscript.

Comment: l. 232 Did the authors mean pathogenic Bacillus, Pseudomonas and Fusarium in this sentence? Fusarium oxysporum, which is also mentioned below, is a generally dangerous phytopathogen as well P. syringae.

Response: We appreciate the Reviewer comment. We are aware that some Bacillus, Pseudomonas and Fusarium are pathogenic, but in this point, we wanted to focus on beneficial effects of Pseudomonas and Bacillus. Some species of Pseudomonas and Bacillus are described to have antagonistic effect against Verticillium dahliae, inhibiting the growth of this fungus. The species of  Pseudomonas and Bacillus were not identified by the authors of the refered study. By this reasons we did not identified the species but only the genera. We apologise because we wrongly included Fusarium in the text (in this sentence), for that reason we removed it. Please see lines 228, 230, 239-242 of the new version of the document.

Comment: l. 240 instead ‘such as’ I would propose ‘having antimicrobial activity’

Response: We correct the text according to the Reviewer suggestion. Please see line 247.

Comments: l. 266 – please improve quality of figure now it is difficult to read

Response: We improved the quality of the figure, please see the new version.

Comment: l. 364, 371 - Does the presence of rhizobia have any significance for the health and growth of olive trees? I ask because these bacteria enter into symbiosis mainly with the roots of legumes and then they can provide the host with N.

Response: The Reviewer is correct, these bacteria enter in symbiosis mainly with the roots of legumes. Nevertheless, the presence of rhizobia can influence indirectly olive health, for example, providing higher nitrogen availability for olive trees. For better clearity, we explain this issue in the manuscript (please see lines 322 to 325 of the new version of the manuscript).

Comment: page 10 - Were the described research results, conducted by different research teams on the microbiota associated with the olive growth environment, carried out with similar parameters for bioinformatic analyses? Such parameters may influence the results of the described microbial communities. If not, in my opinion it should be discussed briefly.

Response: The authors agree with the Reviewer comment, and we confirmed that different methods and bioinformatic analysis were used in the available studies. We included a sentence highlighting the importance of uniformize the methods and analysis (please the section 3.2, lines 301 to 302, and also in the conclusion section).

Comment: l. 388 - on what basis it was indicated that endophytic strains dominated in the rhizosphere. It is true that facultative endophytes can temporarily migrate between plant tissue and soil. However, this sentence is unclear to me.

Response: We agree with the Reviewer and apologize for this incorrectness. We wanted to describe the main bacteria classes in root tissues, rizosphere and bulk soil. We rewrite the text related to this study of Marasco et al. (2021). Please see the new version of the manuscript (lines 369-377).

Comment: l. 508 – ABA - no explanation of the abbreviation

Response: We correct the text according to the Reviewer suggestion (please see lines 533 and 534).

Round 2

Reviewer 1 Report

Comments and Suggestions for Authors

Congratulations, the manuscript has certainly improved.

Thank you for accepting the suggestions proposed.